# Endophytic *Diaporthe* Associated with *Morinda officinalis* in China

**DOI:** 10.3390/jof8080806

**Published:** 2022-07-29

**Authors:** Mei Luo, Wei Guo, Minping Zhao, Ishara S. Manawasinghe, Vladimiro Guarnaccia, Jiawei Liu, Kevin D. Hyde, Zhangyong Dong, Chunping You

**Affiliations:** 1Innovative Institute for Plant Health, Zhongkai University of Agriculture and Engineering, Guangzhou 510225, China; 08luomei@163.com (M.L.); yiququququ@sina.com (W.G.); zmp0609@163.com (M.Z.); 18814383032@163.com (J.L.); kdhyde3@gmail.com (K.D.H.); dongzhangyong@hotmail.com (Z.D.); 2Key Laboratory of Fruit and Vegetable Green Prevention and Control in South-China, Ministry of Agriculture and Rural Affairs, Guangzhou 510225, China; 3Centre for Innovation in the Agro-Environmental Sector, AGROINNOVA, University of Torino, Largo Braccini 2, 10095 Grugliasco, TO, Italy; vladimiro.guarnaccia@unito.it; 4Department of Agricultural, Forestry and Food Sciences (DISAFA), University of Torino, 10095 Grugliasco, TO, Italy; 5Center of Excellence in Fungal Research, Mae Fah Luang University, Chiang Rai 57100, Thailand; 6Deqing Zhongkai Agricultural Technical Innovation Research Co., Ltd., Zhaoqing 526600, China

**Keywords:** new species, new host records, Diaporthales, phylogeny, Chinese traditional medicinal plants, taxonomy

## Abstract

*Diaporthe* species are endophytes, pathogens, and saprobes with a wide host range worldwide. However, little is known about endophytic *Diaporthe* species associated with *Morinda officinalis*. In the present study, 48 endophytic *Diaporthe* isolates were obtained from cultivated *M. officinalis* in Deqing, Guangdong Province, China. The nuclear ribosomal internal transcribed spacer (ITS), partial sequences of translation elongation factor 1-α (*tef1-α*), partial calmodulin (*cal*), histone H3 (*his*), and Beta-tubulin (*β-tubulin*) gene regions were sequenced and employed to construct phylogenetic trees. Based on morphology and combined multigene phylogeny, 12 *Diaporthe* species were identified, including five new species of *Diaporthe longiconidialis*, *D. megabiguttulata*, *D. morindendophytica*, *D. morindae*, and *D. zhaoqingensis*. This is the first report of *Diaporthe chongqingensis*, *D. guangxiensis*, *D. heliconiae*, *D. siamensis*, *D. unshiuensis*, and *D. xishuangbanica* on *M. officinalis*. This study provides the first intensive study of endophytic *Diaporthe* species on *M. officinalis* in China. These results will improve the current knowledge of *Diaporthe* species associated with this traditional medicinal plant. Furthermore, results from this study will help to understand the potential pathogens and biocontrol agents from *M. officinalis* and to develop a disease management platform.

## 1. Introduction

*Diaporthe* Nitschke (1870, (syn. *Phomopsis* (Sacc.) Bubák), (*Diaporthaceae*) [1] includes important plant pathogens, endophytes, and saprobes [2]. Species identification in this genus is based on DNA-based phylogenetic approaches and incorporating sequences from type and voucher specimens to investigate species boundaries. In addition, the incorporation of morphological characters provided a more accurate identification [2]. *Diaporthe* is identified as a cryptic genus and thus morphology alone with phylogeny is challenging to delineate species [3]. Thus, some studies have shown the importance of incorporating additional approaches such as Genealogical Concordance Phylogenetic Species Recognition (GCPSR) and recombination analysis of incorrect species identification in this genus [3]. Even though polyphasic taxonomy has been recently employed for *Diaporthe* systematics, like many fungi, there are no consistent criteria for delineating species.

*Diaporthe* species are widely distributed worldwide and have diverse host associations such as forest trees, vegetables, fruits, and ornamental plants [2,4]. They are compliant and occur as pathogens, endophytes, and saprobes in a wide range of hosts [4]. As plant pathogens, *Diaporthe* species can cause leaf spots, blights, dieback, scab, decay, stem-end rots, trunk, or even wilt diseases including many economically important plants such as Blueberry [5], *Camellia* [6], *Citrus* [7], *Coffea* [8], *Helianthus* [9], *Lithocarpus* [8], *Pyrus* [10], *Senna* [11], and *Vitis* [12]. In addition, *Diaporthe* species are often reported as endophytes [13,14] that live inside the healthy host tissues, without showing any physical symptoms. They were reported as one of the most isolated and dominant endophytic genera, together with *Colletotrichum* and Pestaloid-like, while contributing to the hidden diversity of fungi. [15]. Promputtha et al. [16] isolated 24 *Diaporthe* (as *Phomopsis*) species from 31 morphospecies of sterile *Magnolia liliflora* (*Magnoliaceae*) endophytes. Huang et al. [17] found *Diaporthe* also as one of the dominant endophytes on *Allamanda catharitica*, *Alstonia scholaris*, and other 29 traditional Chinese medicinal plants. 

*Morinda officinalis* F.C. How. (Rutaceae) is a lianoid shrub, which is wildly cultivated in subtropical and tropical areas including China and Vietnam [18,19,20]. In China, this species is cultivated in Guangdong, Guangxi, Fujian, and Hainan Provinces as a traditional medicinal plant. The roots of these plants have been used as tonic or nutrient supplements to treat impotence, osteoporosis, fatigue, rheumatoid arthritis, infertility, and depression [18]. Up until today, several fungal diseases have been reported from *M. officinalis* in China. The black root rot disease caused by *Lasiodiplodia pseudotheobromae* in China [19] and *Fusarium* root rot caused by *Fusarium proliferatum* in Vietnam [20] are mostly reported diseases. However, there are no studies on endophytes associated with *M. officinalis*. 

Following these facts, the objectives of this study were to isolate endophytic *Diaporthe* species associated with healthy *M. officinalis* collected in Deqing city Guangdong province China and to identify species based on multi-locus DNA sequence data. In total, 12 *Diaporthe* species were identified from 48 isolates including five novel species. Species identification and characterisation were performed based on morphology, molecular phylogeny, and pairwise homoplasy index (PHI) analysis. For all identified fungal taxa species, descriptions and illustrations are provided. 

## 2. Materials and Methods

### 2.1. Sampling and Isolation of Endophytic Fungi

Healthy *M. officinalis* leaves, stems, and roots were collected from Deqin city Guangdong province in China (Figure 1). From the collecting site, 10 plants were selected, and from each tree, 10 samples were collected. The collected samples were taken to the laboratory, and endophytic strains were isolated. The samples were washed using running tap water and then sterile water. After that, the samples were cut into 3 × 3 mm segments. Each cutting was surface sterilized in 75% ethanol for 40 s (leaves for 40 s, stems for 60 s, and roots for 90 s), 2.5% sodium hypochlorite for 40 s (leaves for 40 s, stems for 60 s, roots for 70 s), sterile water three times, and then dried on sterile filter paper. The samples were cultured separately on Potato Dextrose Agar (PDA), 2% PDA, Lignocellulose Agar (LCA), Maltose yeast powder medium (M + J), Martin Agar Medium (MD), and P + J medium (100 g healthy and fresh *M. officinalis* tissues boiled for 20 min, filtered with gauze, and fixed the volume to 200 mL. Add 10% volume fraction of decocting juice to PDA medium) plates and incubated at 25 °C following Zhang et al. [21]. Fungi growing at the edges of the tissue were transferred to a new PDA plate. Pure cultures were obtained by following the single mycelium isolation three times [14]. All cultures obtained in this study are deposited in the culture collection of Zhongkai University of Agriculture and Engineering (ZHKUCC). Herbarium materials (as dry cultures) were deposited at Zhongkai University of Agriculture and Engineering (ZHKU).

### 2.2. DNA Extraction and PCR Amplification

For DNA extraction, mycelia were scraped from about seven-day-old cultures on PDA. Total genomic DNA was extracted using the MagPure Plant DNA AS Kit (D6351-AS-06, Guangzhou Magen Biotechnology Co., Ltd., Guangzhou, China) using an Acid purification system (Auto-Pure 32A, Allsheng instruments Co., Ltd., Hangzhou, China). For preliminary species confirmation, the ITS gene region was amplified using the ITS1/ITS4 [22]. The sequences were aligned in GenBank by using the BLAST tool (https://blast.ncbi.nlm.nih.gov/Blast.cgi, accessed on 1 July 2021). Then, four additional gene regions of *tef-1α* [22], *cal* [23], *β-tubulin* [24], and *his* [24,25] were amplified under each PCR conditions (Table 1) and sequenced. PCR amplicons were visualized on 1% agarose electrophoresis gel. The sequencing was performed by Tianyi Huiyuan Biotechnology Co., Ltd., Guangzhou, China. Initial sequence quality was checked using BioEdit 7.25 [26]. The sequence data generated in this study have been deposited in GenBank (Appendix A).

### 2.3. Phylogenetic Analysis

For the phylogenetic analysis, sequences of reference *Diaporthe* species and related taxa were downloaded from NCBI GenBank following [14,27] (Appendix A). Downloaded sequences were aligned together with the sequences obtained in the present study using MAFFT version 7 (http://www.ebi.ac.uk/Tools/msa/mafft/, accessed on 20 January 2022) [28] and adjusted manually using BioEdit 7.25 [26] if necessary. Combined sequence data set of ITS, *tef1-α*, *cal*, *his*, and *β-tubulin* was used following Guarnaccia et al. [9] and Yang et al. [27]. Phylogenetic relationships were inferred using maximum likelihood (ML) in RAxML [29] and Bayesian analyses (BI) in MrBayes (v. 3.0b4) [30]. 

The evolutionary models for each locus used in Bayesian analysis were selected using MrModeltest v. 2.3 [31]. The ML analyses were accomplished using RAxML-HPC2 on XSEDE (8.2.8) [32] in the CIPRES Science Gateway platform [33] using the GTR + I +G model of evolution with 1000 non-parametric bootstrap iterations. Bayesian analysis was performed in MrBayes v. 3.0b4 [30] for the portioned data set. The posterior probabilities (PPs) were determined by Markov chain Monte Carlo sampling (MCMC). Four simultaneous Markov chains were run for 20,000,000 generations, sampling the trees at every 100th generation. From the 200,000 trees obtained, the first 5000 representing the burn-in phase were discarded. The remaining 150,000 trees were used to calculate posterior probabilities (PPs) in a majority rule consensus tree. 

Taxonomic novelties were submitted to the Faces of Fungi database [34] and Index Fungorum (http://www.indexfungorum.org, accessed on 24 May 2022). New species were described following Jayawardena et al. [35] and Manawasinghe et al. [36].

### 2.4. Morphological Characterisation

Agar plugs (5 mm diam.) were taken from the edges of actively growing cultures on PDA and transferred onto the middle of PDA plates and incubated at 25 °C equal hours of alternative dark and fluorescent light for over a month to induce sporulation [13]. Colony characters and pigmentation on PDA were recorded after 4–7 d, 15 d, and 30 d. Colony colours (upper and reverse) were described as in Rayner [37]. Cultures were examined periodically for the development of ascomata and conidiomata. Colony diameters were measured after 3–7 days. The shapes, sizes, and colours of at least 15 pycnidia were recorded with an Eclipse 80i photographic microscope (Nikon, Tokyo, Japan). Pycnidia were cut into 30 µm thin sections by a freezing sliding microtome (Bio-Key science and technology Co., LTD, LEICA CM1860, Weztlar, Germany) for photographing and measuring. Forty conidiophores, alpha conidia, and beta conidia were measured using NISElements BR 3.2, and mean sizes were calculated with their standard deviations (SDs).

### 2.5. Pairwise Homoplasy Index (PHI)

The PHI test was performed using SplitsTree4 v. 4.17.1, [38] to determine the recombination level within closely phylogenetically related species. The concatenated five-locus dataset (ITS + *tef1-α* + *cal* + *his* + *β-tubulin*) was used for the analyses. PHI test results (Fw) > 0.05 indicated no significant recombination within the dataset. The relationships between closely related taxa were visualized in split graphs with both the Log-Det transformation and splits decomposition options.

## 3. Results

### 3.1. Isolation

In total, 48 endophytic *Diaporthe* strains were obtained (eight from roots and 40 from stems). All media used in this study were able to grow *Diaporthe* species. From 48 isolates, 9 isolates were obtained from LCA, 4 isolates were obtained from M + J medium, 10 isolates were obtained from MD, 6 isolates were obtained from CMA, 7 isolates from PDA, 8 were obtained from 2% PDA, and 4 isolates were obtained from P + J medium. 

### 3.2. Phylogenetic Analyses

In the present study, we followed Norphanphoun et al. [39] for the taxonomic treatments of *Diaporthe*. As given in the methods, phylogenetic analyses were arranged in two steps. At first, we developed a genus tree including all species belonging to this genus as given in Guarnaccia et al. [9] and Yang et al. [27]. Then, following Norphanphoun et al. [39], species complexes belonging to isolates from this study were identified, and the final tree was developed. The final analyses were conducted using 271 *Diaporthe* strains (including types strains) with a combined ITS, *tef1-α*, *cal*, *his*, and *β-tubulin* sequence alignment. The phylogenetic tree was rooted in *Diaporthella corylina* (CBS 121124). The final maximum likelihood tree topology was similar to Bayesian analysis. The best scoring RAxML tree with a final likelihood value of −32,402.374230 is given in Figure 2. The matrix consisted of 1065 distinct alignment patterns, with 20.40% of undetermined characters or gaps. Estimated base frequencies were as follows: A = 0.220036, C = 0.307463, G = 0.247268, T = 0.225234; substitution rates AC = 1.189728, AG = 3.654106, AT = 1.419097, CG = 0.937185, CT = 6.230555, GT = 1.000000; gamma distribution shape parameter α = 0.789923. For the Bayesian inference, GTR + I + G model was selected for ITS, TrN + I + G for *tef1-α*, TIM2 + I + G for cal, TIM1 + I + G for his, and TPM3uf + I + G for *β-tubulin*. The Bayesian analyses generated 200,000 trees (average standard deviation of split frequencies: 0.032560), from which 150,000 were sampled after 25% of the trees were discarded as burn-in. The alignment contained 1073 unique site patterns. In this resulted tree, isolates belonging to this study were clustered together with seven known *Diaporthe* species and five novel phylogenetic lineages. They belong to five *Diaporthe* complexes: *D. arecae* (*D. guangxiensis*, *D. eucalyptorum*, *D. xishuangbanica* and *D. zhaoqingensis*), *D. biconispora* (*D. longiconidialis*), *D. eres* (*D. heliconiae*), *D. rudis* (*D. chongqingensis*), and *D. sojae* (*D. megabiguttulata*, *D. morindendophytica*, *D. morindae*, *D. siamensis*, *D. unshiuensis*) species complex. Species descriptions and illustrations of these taxa are given below.

### 3.3. PHI Analysis

In the phylogenetic analysis of *Diaporthe* species, our isolates developed five distinct clades within the genus with significant tree lengths and low statistical support. To confirm species, we conducted PHI analysis for these five clades. There was no evidence of significant genetic recombination (Fw > 0.05) between these novel species of *Diaporthe* and closely related species (Figure 3). These results confirmed that these taxa were significantly different from the existing species of *Diaporthe*.

### 3.4. Taxonomy

***Diaporthe chongqingensis*** Y.S. Guo & G.P. Wang, in Guo, Crous, Bai, et al., Persoonia 45: 146 (2020) (Figure 4).

Index Fungorum number: IF830656

*Endophytic* on *M. officinalis* stems. Sexual morph: not observed. Asexual morph: *Pycnidia* 130–1400 μm × 120–900 μm ($\bar{x}$ = 542 ± 415 μm × 388 ± 267 μm) oblate or subglobose, grey to black, single or multiple cavities, translucent to black conidial drops exuded from the ostioles. *Pycnidia wall* thick, exuding creamy to black conidial droplets from ostioles. *Conidiophores* hyaline, smooth, septate, densely aggregated, cylindrical, straight to sinuous, swelling at the base, tapering towards the apex. *Conidiogenous* cell hyaline, cylindrical, straight, inner wall buds produce sporulation in bottle form. *Alpha conidia* 5–10 μm × 2–3 μm ($\bar{x}$ = 7 ± 0.5 μm × 3 ± 0.2 μm) hyaline, aseptate, fusiform or ellipsoid, biguttulate or multi-guttulate. *Beta* and *gamma conidia* not observed.

Culture characteristics: Colonies on PDA reach 85 mm diam. After 5 days. White cotton flocculent aerial mycelium, while the perimeter edge sparse hyphae, then ochre brown mycelium with several black conidiomata for 15 days and a lot of darker black conidiomata at 30 days. Reverse white and become reddish-brown.

Material examined: China, Guangdong Province, Zhaoqing, isolated from a healthy stem of *M. officinalis*. June 2020, W. Guo (dried culture (ZHKU 22-0032) and living culture (ZHKUCC 22-0043).

Habitat and host: *Pear pyrifolia* [10].

Known distribution: China [10].

Note: A single isolate (ZHKUCC 22-0043) obtained in this study clustered with *D. chongqingensis* (PSCG435) with 37% ML. Morphologically our isolates are similar to *D. chongqingensis* [10]. This species was introduced as a shoot canker pathogen on pear (*Pear pyrifolia*) in China [10]. This is the first report of *D. chongqingensis* as an endophyte on the *M. officinalis* stem.

***Diaporthe eucalyptorum*** Crous & R.G. Shivas, in Crous, Summerell, Shivas, et al., Persoonia 28: 153 (2012) (Figure 5). 

Index Fungorum number: IF800374

*Endophytic* on *M. officinalis* stem. Sexual morph: not observed. Asexual morph: *Pycnidia* on PDA 360–770 μm × 330–600 μm ($\bar{x}$ = 573 ± 128 μm × 478 ± 81 µm) in diam., black, erumpent; white to black conidial droplets exuding from central ostioles. *Pycnidia wall* with black pseudoparenchymatous, exuding transparent to black conidial droplets from ostioles. *Conidiophores* hyaline, smooth, densely aggregated, cylindrical, straight to sinuous, swelling at the base, tapering towards the apex. *Conidiogenous* cell phialidic, cylindrical, terminal and lateral, with slight taper towards apex. *Alpha conidia* 5–10 × 2–4 µm ($\bar{x}$ = 5 ± 0.6 × 3 ± 0.4 µm), aseptate, hyaline, smooth, guttulate, fusoid, tapering towards ends, straight, apex subobtuse, base subtruncate. *Beta conidia* 15–30 × 1–3 µm ($\bar{x}$ = 20 ± 3 μm × 2 ± 0.4 µm), aseptate, hyaline, hamate, filiform, tapering toward both ends.

Culture characteristics: Colonies on PDA reach 85 mm diam. after 4 days. Cottony and radially with abundant aerial mycelium, sparse in the margin, white on surface, then turn to grey. Reverse white to pale yellow.

Material examined: China, Guangdong Province, Zhaoqing, isolated from a healthy stem of *M. officinalis*. June 2020, W. Guo, (dry culture (ZHKU 22-0033), and living culture (ZHKUCC 22-0044, 22-0045).

Habitat and host: *Eucalyptus* sp. [40].

Known distribution: Australia [40].

Note: In the multigene phylogenetic tree, two isolates (ZHKUCC 22-0044 and ZHKUCC 22-0045) from this study clustered together with the *D. eucalyptorum* (CBS 132525) with 85% ML and 1.00 BYPP values. Morphologically, the ZHKUCC 22-0044 isolate is similar to those in the original description of *D. eucalyptorum* [40]. The ZHKUCC 22-0044 strain developed both *alpha* and *beta conidia* while the *D. eucalyptorum* type strain (CBS 132525) develops *alpha conidia* only. *Diaporthe eucalyptorum* was introduced from diseased leaves of *Eucalyptus* sp. in Australia [40]. This is the first report of *D. eucalyptorum* as an endophyte on *M. officinalis*.

***Diaporthe guangxiensis*** Dissanayake, X.H. Li & K.D. Hyde, in Manawasinghe, Dissanayake, Li, et al., Frontiers in Microbiology 10 (no. 1936): 14 (2019) (Figure 6). 

Index Fungorum number: IF552578

*Endophytic* on *M. officinalis* stem. Sexual morph: not observed. Asexual morph: *Pycnidia* on PDA 260–670 μm × 200–580 μm ($\bar{x}$ = 465 ± 92 μm × 412 ± 91 µm) in diam., superficial, scattered on PDA, dark brown to black, globose, solitary, or clustered in groups. *Pycnidia walls* pseudoparenchymatous, conidial cirrus extruding from ostiole appearing as white to black droplets. *Conidiophores* hyaline, smooth, densely aggregated, unbranched. *Beta conidia* 10–30 × 1–2 µm ($\bar{x}$ = 16 ± 5 μm × 1 ± 0.5 µm) aseptate, hyaline, hamate, filiform, guttulate, tapering toward both ends. *Alpha* and gamma *conidia* not found.

Culture characteristics: Colonies on PDA reach 85 mm diam. after 25 days. White aerial mycelia and then turn to yellowish-brown. Reverse first white, then dark brownish-green.

Material examined: China, Guangdong Province, Zhaoqing, isolated from a healthy stem of *M. officinalis*. June 2020, W. Guo, dried culture, ZHKU 22-0034, and living culture (ZHKUCC 22-0046)

Habitat and host: *Vitis vinifera* [12].

Known distribution: China [12].

Note: In the phylogenetic analyses, a single isolate (ZHKUCC 22-0046) from this study clustered together with the *D. guangxiensis* strain (JZB320094), 76% ML and 1.0 BYPP values. Morphologically, the present isolate is similar to those in the original description of *D. guangxiensis* [12]. *Diaporthe guangxiensis* was introduced from diseased *V. vinifera* trunks [12]. The type strain (JZB320094) of *D. guangxiensis* developed both alpha and beta conidia while the ZHKUCC 22-0046 strain develop beta conidia only. This is the first report of *D. guangxiensis* as an endophyte on *M. officinalis*.

***Diaporthe heliconiae*** S.T. Huang, J.W. Xia, X.G. Zhang & Z. Li, in Sun, Huang, Xia, et al., MycoKeys 77: 76 (2021) (Figure 7). 

Index Fungorum number: IF 837592

*Endophytic* on *M. officinalis* stem. Sexual morph: not observed. Asexual morph: *Pycnidia* 400–1680 μm × 230–1620 μm ($\bar{x}$ = 1268 ± 325 μm × 1152 ± 354 µm), solitary or aggregated in groups, erumpent, thin-walled, superficial to embedded on PDA, dark brown to black, globose or subglobose, exuding transparent to dark brown conidial cirrus from the ostioles. *Pycnidia wall* pseudoparenchymatous, conidial cirrus extruding from ostiole appearing as transparent droplets. *Conidiophores* hyaline, smooth, densely aggregated, unbranched. *Conidiogenous* cell 15–40 × 1–3 µm ($\bar{x}$ = 27 ± 5 μm × 2 ± 0.4 μm), hyaline, aseptate, cylindrical, straight to sinuous, branched. *Alpha conidia* 5–10 × 2–5 µm ($\bar{x}$ = 6 ± 0.7 μm × 3 ± 0.4 μm), hyaline, smooth, aseptate, ellipsoidal, 2–4 guttulate, apex subobtuse, base subtruncate. *Beta conidia* 20–30 × 1–2 µm ($\bar{x}$ = 26 ± 3 μm × 2 ± 0.3 μm), hyaline, aseptate, filiform, mostly straight and sometimes slightly curved, tapering towards the apex. *Gamma conidia* not observed.

Culture characteristics: Colonies on PDA reach 85 mm diam. after 15 days. Aerial mycelium abundant, cottony, white, dense in the center, sparse near the margin. White on surface side, then yellowish-brown; reverse white to brown.

Material examined: China, Guangdong Province, Zhaoqing, isolated from a healthy stem of *Morinda officinalis*. June 2020, W. Guo (dried culture (ZHKU 22-0035), and living culture (ZHKUCC 22-0047, 22-0048).

Habitat and host: *Heliconia metallica* [41].

Known distribution: China [41].

Note: In the multigene phylogenetic analysis, two isolates (ZHKUCC 22-0047 and ZHKUCC 22-0048) from the present study clustered together with the *D. heliconiae* (SAUCC194.77) with 100% ML and 1.00 BYPP values. Morphologically, the present isolate is similar to those in the original description of *D. heliconiae* [41]. *Diaporthe heliconiae* was introduced as a pathogen on *H. metallica* petiole [41]. This is the first report of *D. heliconiae* as an endophyte on *M. officinalis*.

***Diaporthe longiconidialis*** M. Luo, W. Guo, Manawas., M. P. Zhao, K. D. Hyde & C. P. You, sp. nov. (Figure 8). 

Index Fungorum number: IF553402

Etymology: In reference to its long spores.

Holotype: ZHKUCC 22-0040

*Endophytic* on *M. officinalis* root and stem. Sexual morph: not observed. Asexual morph: *Pycnidia* 40–400 × 30–160 μm ($\bar{x}$ = 150 ± 100 μm × 70 ± 40 μm), subglobose, flask or irregularly shaped, single or multiple cavities. *Pycnidial wall* consisting of sevearal layers of medium transparent textura globosa-angularis. *Conidiophores* hyaline, unbranched, densely aggregated, cylindrical-filiform, straight to sinuous, 1-septate. *Conidiogenous* cell phialidic, cylindrical, terminal and lateral, with slight taper towards apex. *Alpha conidia* 6–10 × 2–4 μm ($\bar{x}$ = 8 ± 1 μm × 3 ± 0.3 μm), hyaline, fusiform, ellipsoid, long ellipsoid or lanceolate, contains at least one guttulate. *Beta conidia* 15–30 × 1–3 μm ($\bar{x}$ = 20 ± 3 μm × 2 ± 0.3 μm), hyaline, linear, curved.

Culture characteristics: Colonies on PDA reach 85 mm diam. after 5 days. White cotton with some polygon mycelium as stars in the center, then with purple pigmentation. Fruiting body occurred from about 15 days, grey to dark black with orange surroundings. Reverse white with some purple pigmentation.

Material examined: China, Guangdong Province, Zhaoqing, isolated from a healthy root of *M. officinalis*. June 2020, W. Guo, dried culture (ZHKU 22-0040), and living culture (ZHKUCC 22-0058, ZHKUCC 22-0059–0066).

Habitat and host: Healthy stem and root of *M. officinalis*.

Known distribution: China (Zhaoqing, Guangdong Province).

Note: In the polygenic analysis, eight isolates from the present study clustered together with the *Diaporthe biconispora* and *Diaporthe pometiae* ex-type strain with 100% ML and 1.00 BYPP values. Morphologically, *Alpha conidia* of strain ZHKUCC 22-0058 ($\bar{x}$ = 8 ± 1 × 3 ± 0.3 μm) were similar to those in the original description of *D. biconispora* [13]. The colony morphology of our isolates was different to those in the original description of *D. biconispora,* the surface of the colony is white and light yellow, the center of the back is black, and the sides are light yellow [13]. *Alpha conidia* of *D. pometiae* ($\bar{x}$ = 6.7 × 3.1 μm) were smaller than the ZHKUCC 22-0058 strain. The colony morphology of our isolate was different to those in the original description of *D. pometiae*. The surface of the colony is white, and the back is white to light grey [42]. *Diaporthe biconispora* has 3% nucleotide differences in ITS (497 nucleotides), 3% differences in *tub2* (353 nucleotides), 4% differences and 4% gaps in *tef1-α* (355 nucleotides). *Diaporthe pometiae* has 5% differences and 3% gaps in ITS (541 nucleotides), 6% differences, and 1% gaps in *tub2* (471 nucleotides). In addition, there is no evidence of significant genetic recombination (Fw = 0.300) in the PHI analysis. Considering both morphological and molecular data, these isolates were identified as a new species. 

***Diaporthe******megabiguttulata*** M. Luo, W. Guo, Manawas., M. P. Zhao, K. D. Hyde, & C. P. You, sp. nov. (Figure 9).

Index Fungorum number: IF553404

Etymology: In reference to its spores with two big guttulates.

Holotype: ZHKUCC 22-0067

Description: *Endophytic* on *M. officinalis* root. Sexual morph: not observed. Asexual morph: *Pycnidia* 50–440 × 40–270 μm ($\bar{x}$ = 180 ± 120 × 120 ± 70 μm), oblate or hemispherical, single, or multiple cavity or rotary cavity. *Pycnidial wall* consisting of several layers of medium transparent textura globosa-angularis. *Conidiogenus* cells hyaline, phialidic. *Conidiogenous* cell hyaline, phialidic. *Alpha conidia* 5–10 × 2–3 μm ($\bar{x}$ = 6 ± 0.5 × 20 ± 0.2 μm), hyaline, ellipsoid or lanceolate, biguttulate, with one end obtuse and the other acute. *Beta conidia* 20–30 × 1–2 μm ($\bar{x}$ = 30 ± 3 × 1 ± 0.3 μm), hyaline, hamate or curved, base truncate.

Culture characteristics: Colonies on PDA reach 85 mm diam. after 4 days. Aerial mycelium, white or pale white, with fruiting body after 20 days. Reverse white, then turns to pale.

Material examined: China, Guangdong Province, Zhaoqing, isolated from a healthy stem of *M. officinalis*. June 2020, W. Guo, dried culture (ZHKU 22-0041), and living culture (ZHKUCC 22-0067 ex-type and ZHKUCC 22-0068 ex-paratype).

Habitat and host: Healthy stem and root of *M. officinalis*.

Known distribution: China (Zhaoqing, Guangdong Province).

Note: In the polygenic phylogenetic tree, two isolates (ZHKUCC 22-0067 and 22-0068) from the present study clustered together with the *D. unshiuensis* [13] and *D. longicolla* [42] ex-type strain, 99% ML, and 1.00 BYPP values. Morphologically, alpha conidia of *D. unshiuensis* ($\bar{x}$ = 6.5 ± 0.6 × 2.8 ± 0.4 µm) [13] were wider than the ZHKUCC 22-0067 ($\bar{x}$ = 6 ± 0.5 × 20 ± 0.2 μm), the colony gradually turns gray on the front and dark gray in the middle on the back. Alpha conidia of *D. longicolla* ($\bar{x}$ = 6.4 ± 0.3 × 2.3 ± 0.1 μm) [43] were longer, and both of them were not found beta conidia. Compared between *D. unshiuensis*, *D. longicolla* and strain ZHKUCC 22-0067, *D. unshiuensis* has 1% differences in ITS (491 nucleotides), while *D. longicolla* has 4% differences and 2% gaps in ITS (550 nucleotides). For the *tub2* gene, there are 1% differences (446 nucleotides) with *D. longicolla* and 1% differences (373 nucleotides) with *D. unshiuensis*. In addition, there is no evidence of significant genetic recombination (Fw = 1.000) in the PHI analysis. Considering both morphological and molecular data, these isolates were identified as novel species. 

***Diaporthe morindae*** M. Luo, W. Guo, M. P. Zhao, Manawas., K. D. Hyde & C. P. You, sp. nov. (Figure 10).

Index Fungorum number: IF553409

Etymology: In reference to its host of *M. officinalis*.

Holotype: ZHKUCC 22-0072

*Endophytic* on *Morinda officinalis* root and stem. Sexual morph: not observed. Asexual morph: *Pycnidia* 50–380 × 30–160 μm ($\bar{x}$ = 170 ± 90 μm × 90 ± 40 μm), oblate, subglobose, flask or irregularly shaped, multiple cavity or rotary cavity. *Pycnidial wall* consisting of sevearal layers of medium transparent textura globosa-angularis. *Conidiogenus* cells hyaline, phialidic. *Conidiogenous* cell hyaline, phialidic. *Alpha conidia* 6–7 × 2–4 μm ($\bar{x}$ = 6 ± 0.3 μm × 3 ± 0.3 μm), hyaline, ellipsoid or torque circular, blunt at ends, mono- or biguttulate. *Beta* and *gamma conidia* not observed.

Culture Characteristics: Colonies on PDA reach 85 mm diam. after 5 days. White cotton flocculent aerial hyphae. At 15 days, the central part with a diameter of 10 mm was black on the back of the colony, and the part beyond the diameter of 40 mm began to turn black, then the black deepened, and oil droplets appeared. At 30 days, black dots appeared in the colony, and the fruiting body was gradually produced.

Material examined: China, Guangdong Province, Zhaoqing, isolated from a healthy stem of *M. officinalis*. June 2020, W. Guo, dried culture (ZHKU 22-0043), and living culture (ZHKUCC 22-0072 ex-type and ZHKUCC 22-0073–0076).

Habitat and host: healthy stem and root of *M. officinalis*.

*Known distribution*: China (Zhaoqing, Guangdong Province).

Note: In the polygenic phylogenetic tree five isolates in the present study clustered together with the *D. hubeiensis* (JZB320121 and JZB320122) and *D. tectoane* (MFLUCC 14.1139, MFLUCC 12.0777, and MFLUCC 14.1138) strains with 86% ML and 0.99 BYPP. Morphologically, *Alpha conidia* of *D. hubeiensis* ($\bar{x}$ = 6.1 × 1.8 μm) [12] and *D. tectonae* ($\bar{x}$ = 5.5 × 2.6 μm) [43] are smaller. In pairwise nucleotide comparisons, *Diaporthe hubeiensis* has 19% differences and 18% gaps in ITS (543 nucleotides), 2% differences in *tub2* (522 nucleotides), 10% differences and 7% gaps in *cal* (510 nucleotides), and 5% differences and 4% gaps in *tef1-ɑ* (339 nucleotides). *D. tectonae* has 3% differences and 2% gaps in ITS (547 nucleotides), 7% differences and 4% gaps in *tub2* (485 nucleotides), 3% differences and 1% gaps in *cal* (476 nucleotides), and 2% differences in *tef1-ɑ* (315 nucleotides). In addition, there is no evidence of significant genetic recombination (Fw = 0.238) in the PHI analysis. Considering both morphological and molecular data, these isolates were identified as a novel species. 

***Diaporthe******morindendophytica*** M. Luo, W. Guo, Manawas., M. P. Zhao, K. D. Hyde, & C. P. You, sp. nov. (Figure 11). 

Index Fungorum number: IF553423

Etymology: In reference to its endophytic nature in the host of *Morinda officinalis*.

Holotype: ZHKUCC 22-0069

*Endophytic* on *M. officinalis*. Sexual morph: not observed. Asexual morph: *Pycnidia* 40–200 × 30–130 μm ($\bar{x}$ = 120 ± 450 μm × 70 ± 20 μm), oblate, subglobose or irregularly shaped, single or multiple cavities. *Pycnidial wall* consisting of sevearal layers of medium transparent textura globosa-angularis. Conidial masses produced as black droplets extruding through the ostioles. *Conidiogenus* cells hyaline, phialidic. *Alpha conidia* 5–10 × 2–3 μm ($\bar{x}$ = 6 ± 0.5 μm × 3 ± 0.3 μm), hyaline, ellipsoid, torque circular or lanceolate, blunt at both ends or slightly pointed at one end, biguttulate. *Beta conidia* 20–30 × 1–2 μm ($\bar{x}$ = 20 ± 3 μm × 2 ± 0.2 μm), hyaline, linear, and curved.

Culture characteristics: Colonies on PDA reach 85 mm diam. after 4 days. White cotton flocculent aerial hyphae, turned grayish-black or yellow. Reverse yellowish-brown, with a large number of black dots.

Material examined: China, Guangdong Province, Zhaoqing, isolated from a healthy stem of *M. officinalis*. June 2020, W. Guo, dried culture (ZHKU 22-0042), and living culture (ZHKUCC 22-0069 ex-type and ZHKUCC 22-0070, 22-0071 ex-paratype).

Habitat and host: Healthy stem of *M. officinalis*.

Known distribution: China (Zhaoqing, Guangdong Province).

Note: In the polygenic phylogenetic tree three isolates (ZHKUCC 22-0069–0071) from the present study cluster together with the *Diaporthe tectonendophytica* (MFKUCC 13.0471 and SAUCC0463), 100% ML and 1.00 BYPP values. Morphologically, *D. tectonendophytica* develops bigger pycnidia ($\bar{x}$ = 542 × 660 μm) [3] than our isolates. *Alpha conidia* ($\bar{x}$ = 5 × 2.2 μm) and *Beta conidia* ($\bar{x}$ = 23 × 1.3 μm) are smaller than the ZHKUCC 22-0069 isolate. The culture characteristics are different. *Diaporthe tectonendophytica* was first white and turn to yellowish grey, while the ZHKUCC 22-0069 strain turned grayish-black or yellow. There are 3% differences in ITS (547 nucleotides), 3% differences in *tub2* (487 nucleotides), 3% differences in *tef1-α* (307 nucleotides) between *D. tectonendophytica* and strain ZHKUCC 22-0069. In addition, there is no evidence of significant genetic recombination (Fw = 0.205) in the PHI analysis. Considering both morphological and molecular data, these isolates were identified as novel species. 

***Diaporthe siamensis*** Udayanga, Xing Z. Liu & K.D. Hyde, in Udayanga, Liu, Mckenzie et al., Cryptog. Mycol. 33(3): 298 (2012) (Figure 12). 

Index Fungorum number: IF800826

*Endophytic* on *M. officinalis*. Sexual morph: not observed. Asexual morph: *Pycnidia* 100–700 μm × 40–500 μm ($\bar{x}$ = 320 ± 160 μm × 220 ± 130 μm) subglobose, flask or irregularly shaped, single or multiple cavities. *Pycnidia wall* pseudoparenchymatous, conidial cirrus extruding from ostiole appearing as transparent to yellowish-brown droplets. *Conidiophores* cylindrical, wide at the base, hyaline, simple, densely aggregated. *Conidiogenus* cells hyaline, phialidic, cylindrical. *Alpha conidia* 5–10 × 2–3 μm ($\bar{x}$ = 7 ± 0.4 μm × 3 ± 0.3 μm), hyaline, fusiform, ellipsoid, torque circular or lanceolate, blunt at both ends or slightly pointed at one end, mostly biguttulate. *Beta conidia* 20–40 × 1–2 μm ($\bar{x}$ = 30 ± 5 μm × 2 ± 0.2 μm), hyaline, linear, curved.

Culture characteristics: Colonies on PDA reach 85 mm diam. after 5 days. White cotton flocculent aerial mycelium in the center, surrounding sparse hyphae, and then become yellowish brown, margin lobate. Reverse white and then purple grey due to pigment formation.

Material examined: China, Guangdong Province, Zhaoqing, isolated from a healthy stem of *M. officinalis*. June 2020, W. Guo, dried culture (ZHKU 22-0036), and living culture (ZHKUCC 22-0049 and ZHKUCC 22-0050).

Habitat and host: *Dasymaschalon trichophorum* [13,43], *Pandanus tectorius* [44].

Known distribution: China [13,43], Thailand [44].

Note: In the phylogenetic analysis, two isolates (ZHKUCC 22-0049 and ZHKUCC 22-0050) obtained in this study clustered with *Diaporthe siamensis* (MFLUCC 17.0591 and MFLUCC 10.0573a) with 100% ML and 1.00 BYPP values. The ZHKUCC 22-0049 isolate from this study is morphologically similar to the *D. siamensis* type description [3] by conidia with similar shapes and dimensions. This species was first reported on necrotic leaves of *Dasymaschalon* in China by Udayanga et al. [3]. This is the first report of *D. siamensis* as an endophyte on *M. officinalis*.

***Diaporthe unshiuensis*** F. Huang, K.D. Hyde & Hong Y. Li, in Huang, Udayanga, Mei, et al., Fungal Biology 119(5): 344 (2015) (Figure 13). 

Index Fungorum number: IF810845

*Endophytic* on *M. officinalis* stem. Sexual morph: not observed. Asexual morph: *Pycnidia* 350 × 300 μm, globose, subglobose or irregular, bark brown, pseudoparenchymatous walls. Conidial masses produced as black droplets extruding through the ostioles. *Pycnidia wall* consisting of several layers of medium transparent *textura globosa-angularis. Pycnidia wall* consisting of several layers of medium transparent *textura globosa-angularis*, conidial cirrus extruding from ostiole appearing as black droplets. *Conidiophores* cylindrical, wide at the base, hyaline, simple. *Alpha conidia* 5–10 × 2–3 μm ($\bar{x}$ = 6 ± 0.8 × 3 ± 0.3 μm), ellipsoidal or clavate, base truncate, aseptate, smooth, biguttulate. *Beta conidia* 20–30 × 1–3 μm ($\bar{x}$ = 25 ± 3 × 2 ± 0.4 μm), hyaline, linear, curved, hook-shaped. 

Culture characteristics: Colonies on PDA reach 85 mm diam. after 9 days. Surface white and turning to grey with ageing, reverse white to grey with dark grey at the centre.

Material examined: China, Guangdong Province, Zhaoqing, isolated from healthy root of *M. officinalis*. June 2020, W. Guo, dried culture (ZHKU 22-0037), and living culture (ZHKUCC 22-0051–53).

Habitat and host: *Carya illinoinensis*, *Citrus* sp., *Citrus unshiu*, *Fortunella margarita*, *Glycine max* and *Vitis vinifera* [45].

Known distribution: China, Louisiana [45].

Note: In the multigene phylogenetic analysis, three isolates (ZHKUCC 22-0051–53) from the present study cluster together with the *D. unshiuensis* strains (ZJUD49 and ZJUD50), 87% ML and 1.00 BYPP values. Morphologically, the ZHKUCC 22-0051 isolate is similar to those in the original description of *D. unshiuensis* [13]. *Diaporthe unshiuensis* was introduced as an endophyte on the *Citrus unshiu* fruits, on branches and twigs of *F. margarita* [13]. Our strains develop both *alpha* and *beta* conidia while the *D. unshiuensis* type strain (ZJUD52) develops only *alpha* conidia [13]. This is the first report of *D. unshiuensis* as an endophyte on *M. officinalis*.

***Diaporthe xishuangbanica*** Y.H. Gao & L. Cai, in Gao, Liu, Duan, et al., IMA Fungus 8(1): 179 (2017) (Figure 14).

Index Fungorum number: IF820685

*Endophytic* on *M. officinalis* root and stem. Sexual morph: not observed. Asexual morph: *Pycnidia* 30–360 × 20–230 μm ($\bar{x}$ = 119 ± 90 μm × 72 ± 60 μm), oblate, subglobose or irregularly shaped, single or multiple cavities. Conidial masses produced as transparent droplets extruding through the ostioles. *Pycnidia wall* consisting of several layers of medium transparent *textura globosa-angularis*. Conidial masses produced as transparent droplets extruding through the ostioles. *Conidiophores* hyaline, branched, septate. *Conidiogenus* cells grow from the conidia, hyaline, phialidic. *Beta conidia* 20–30 × 1–2 μm ($\bar{x}$ = 27 ± 3 μm × 2 ± 0.3 μm), hyaline, linear, and curved. *Alpha conidia* and *gamma conidia* not observed.

Culture characteristics: Colonies on PDA reach 85 mm diam. after 5 days. White cotton flocculent aerial mycelium in the center, surrounding sparse hyphae. At 15 days, white cotton flocculent aerial hyphae and a few black dots appear, reddish pigments are produced on the back side of the colony. At 30 days, the black dots increased, and the fruiting body was produced.

Material examined: China, Guangdong Province, Zhaoqing, isolated from healthy root of *M. officinalis*. June 2020, W. Guo, dried culture (ZHKU 22-0038), and living culture (ZHKUCC 22-0054, 22-0055).

Habitat and host: *Camellia sinensis* [8].

Known distribution: China [8].

Note: In the multigene phylogenetic tree two isolates (ZHKUCC 22-0054, 22-0055) from the present study cluster together with the *D. xishuangbanica* strains (LC9707 and CGMCC 3.18282) with 99% ML and 1.0 BYPP. Morphologically, the ZHKUCC 22-0054 isolate is similar to those in the original description of *D. xishuangbanica* [8]. However, *D. xishuangbanica* developed *alpha* conidia while the ZHKUCC 22-0054 isolate developed only *beta* conidia. *Diaporthe xishuangbanica* was introduced from *Camellia sinensis* diseased leaves. This is the first report of *D. xishuangbanica* as an endophyte on *M. officinalis*.

***Diaporthe zhaoqingensis*** M. Luo, M. P. Zhao, W. Guo, Manawas., K. D. Hyde & C. P. You, **sp. nov**. (Figure 15).

Index Fungorum number: IF553427

Etymology: In reference to the Zhaoqing city, Guangdong Province, from where the samples were collected.

Holotype: ZHKUCC 22-0039

*Endophytic* on *M. officinalis* root and stem. Sexual morph: not observed. Asexual morph: *Pycnidia* 200–1000 × 170–1000 μm ($\bar{x}$ = 574 ± 272 × 520 ± 277 μm), subglobose, flask or irregularly shaped, single or multiple cavities. *Pycnidial* wall consisting of several layers of medium transparent *textura globosa-angularis. Conidiophores* hyaline, unbranched, densely aggregated, cylindrical-filiform, straight to sinuous. *Conidiogenus* cells phialidic, cylindrical, terminal, and lateral, with slight taper towards apex. *Gamma conidia* 10–30 × 0.5–2 μm ($\bar{x}$ = 15 ± 4 μm × 2 ± 0.5 μm), fusiform, hyaline. Alpha and *Beta conidia* not observed.

Culture characteristics: Colonies on PDA reach 50 mm diam. After 7 days. White cotton flocculent aerial mycelium in the center, surrounding sparse hyphae, with lobate margin. First white, then turn to grey and yellow grey.

Material examined: China, Guangdong Province, Zhaoqing, isolated from healthy stem and root of *Morinda officinalis*. June 2020, W. Guo, dried culture ZHKU 22-0039, and living culture ZHKUCC 22-0039 and 22-0040 ex-type.

Habitat and host: Healthy stem and root of *M. officinalis*.

Known distribution: China (Zhaoqing, Guangdong Province).

Note: In the polygenic phylogenetic tree two isolates (ZHKUCC 22-0039 and 22-0040) from the present study clustered as an independent clade with 100% ML and 1.00 BYPP support. In the PHI analysis, there was no evidence of significant genetic recombination (Fw = 0.550) among our isolates (ZHKUCC 22-0039) and its closely related taxa (*D. australiana* (BRIP 66145), *D. eucalyptorum* (CBS 132525), *D. hongkongensis* (CBS 115448), *D. lithocarpus* (CGMCC 3.15175), and *D. salinicola* (MFLUCC 18.0553)). Morphologically, ZHKUCC 22-0039 strain develops a significantly higher amount of gamma conidia while other closely related species *D. australiana* [46], *D. eucalyptorum* [41], *D. hongkongensis* [2], *D. lithocarpus* [11], and *D. salinicola* [47] are not developing. Considering both morphological and molecular data, here we introduce our isolates as a novel *Diaporthe* species.

## 4. Discussion

Endophytic fungi live inside the healthy host tissues [48,49]. They are a common and diverse group of fungi [12,49]. *Diaporthe* species are ubiquitous endophytes on numerous hosts. Skaltsas et al. [50] isolated endophytic *Diaporthe* from *Hevea brasiliensis*, *H. guianensis*, and *Micandra* spp. in Cameroon, Mexico. Rhoden [51] isolated 97 strains from *Trichilia elegans* (*Meliaceae*) in Brazil, while the *Diaporthe* were the most frequently isolated genus. This reflects the species richness of *Diaporthe* species as endophytes in many hosts. In the present study, 48 endophytic *Diaporthe* isolates were obtained from *M. officinalis* in China from stems and roots. Based on the multigene phylogeny, all 48 isolates from this study were grouped in 12 distinct clades within the *Diaporthe* phylogenetic tree. Among them, seven new host records were identified: *Diaporthe chongqingensis*, *D. guangxiensis*, *D. heliconiae*, *D. siamensis*, *D. unshiuensis*, and *D. xishuangbanica*. The remaining five species were identified as novel species: *Diaporthe longiconidialis, D. megabiguttulata, D. morindendophytica, D. morindae, and D. zhaoqingensis*. Our study is the first comprehensive study on endophytic fungi associated with *M. officinalis* in China.

During the last few years, many *Diaporthe* species were introduced. For example, in 2020, 34 new species were introduced, and in 2021, 37 new species were introduced (Index Fungorum 2022, accessed on 15 June 2022). Almost all these species were introduced based on molecular phylogeny following the phylogenetic species concept [13,27,46]. However, *Diaporthe* species do not have enough morphologies to distinguish; thus, the phylogenetic tree with five loci became the key tool to define the species. This resulted in over 200 species which are genetically closer and define *Diaporthe* as cryptic species. What is missing here is that there are no discussions on the different species or different genotypes of the existing species. In the phylogenetic analysis of the present study, *D. morindae*, a new species from our study, was clustered together with *D. hubeiensis*, *D. tectonae*, and *D. tulliensis*. These four species develop a distinct clade from other *Diaporthe* species in the tree. Within this clade, these species develop independent lineages, yet their morphologies are overlapping. However, it is quite interesting as this clustering might be based on the geography and host in which, *D. morindae* and *D. hubeiensis* introduced from China are associated with Citrus (this study) and *Vitis* [12]. *Diaporthe tectonae* was introduced from m *Tectona grandis* from Thailand [52]. In contrast, *D. tulliensis* was introduced from the rotted stem end of the fruit of *Theobroma cacao* in Australia [53].

Additional analyses such as recombination tests and phylogenetic incompatibility are employed, but the species delineation enigma of *Diaporthe* is unresolved. Therefore, in the present study, we followed polyphasic taxonomic approaches which included phylogeny, morphology, and recombination analysis to identify the species. For existing species, species definition was based on phylogeny and morphology. Furthermore, we emphasise that taxon sampling is critical to establishing a primary tree to void the possible false introduction of species. However, it is necessary to discuss “what is the species” in *Diaporthe*.

Medicinal plants have a high economical and important role in human cultures worldwide. *Morinda officinalis* is a famous traditional medicinal plant, which has various biological effects such as anti-inflammatory activities [53], antiosteoporotic [54], antifatigue [55], anti-rheumatoid arthritis [56], and anti-oxidant [57]. It has been mentioned that endophytes live inside the host might affect the phenotype of the host in many ways, including providing resistance to pathogens [58,59], promoting seed germination or/and plant growth [60], herbivores [61], weed control [62,63], resistance to abiotic stresses, or even in litter decomposition [64]. In addition, some studies have revealed that secondary metabolites produced by endophytic fungi could be a novel source of medicinal compounds [64]. Though two new metabolites from *M. officinalis* endophytic *Alternaria* sp. A744 [63] and two polyketide compounds from the *M. officinalis* endophytic *Trichoderma spirale* A725 [65] were found. However, little is known about the endophytic fungi of *M. officinalis*. Therefore, further studies are necessary to explore the relationship between medicinal properties and a plant’s endophytic biota.

One endophytic species might occur in different tissues in the same host [13]. In the present study, endophytes *Diaporthe* were successfully isolated from *M. officinalis*’s roots and stems. *Diaporthe evcalyptorum*, *D. longiconidialis*, *D. morindae* and *D. xishuangbanica* were isolated from roots and stems of *M. officinalis*, while other species were isolated from only one tissue type. However, we were not able to isolate any *Diaporthe* species from leaves. Gond et al. [66] obtained only two endophytic *Diaporthe* strains from leaves of *Aegle marmelos* Correae (*Rutaceae*). Dong et al. [14] identified two endophytic *Diaporthe* species from *Citrus grandis* cv. Tomentosa while 22 strains were from fruits or twigs. Huang et al. [13] did not find any endophytes *Diaporthe* from *Citrus* leaves. These variations might be a result of differences in the tissue organization structure and the different nutrition content of each tissue type [13,67]. However, the exact underlying reasons and mechanisms for these variations are not known. At the same time, further studies are needed to compare the variations in endophytic colonization according to different seasons or different stages of maturity of the plant.

Optimization of cultural conditions is the most common and simple method to obtain different fungal diversity. Zhou et al. [68] found that the addition of vitamins in the media significantly increased the diversities of isolated fungi, and the increment reached 207% and 81% at the generic level at both 4 °C and 25 °C, respectively, conditions. In the present study, *D. morindendophytica* was isolated from three media (MD, CMA, and P++J medium), while two species of them including *D. longiconidialis*, and *D. morindae* were isolated from all the media tested (PDA, 2% PDA, LCA, M + J, MD and P + J medium). In addition, these four species were isolated from both roots and stems. Therefore, these species might be the dominant endophytic *Diaporthe* species in *M. officinalis*. Even though *Diaporthe* species can grow in different media, *D. biguttulata*, *D. chongqingensis*, *D. heliconiae*, *D. heterophyllae*, *D. guangxiensis*, and *D. megabiguttulata* were only isolated in one medium. Thus, optimizing and employing several media to isolate endophytes will enhance the diversity of endophytes obtained from a particular host.

Several endophytes such as *Colletotrichum* and *Botryosphaeriaceae* have been considered latent pathogens or opportunistic pathogens [69,70]. *Diaporthe* is also known to be opportunistic pathogens. Huang et al. [13] observed that some *Diaporthe* species associated with *Citrus* in China may act as opportunistic pathogens. Dong et al. [14] found *Diaporthe limonicola* as endophyte in *C. grandis* cv. Tomentosa, while it has also been reported as pathogenic on *Citrus* sp. In China [13] and as a dieback pathogen of lemon trees in Europe [7]. In the present study, all species isolated are known pathogenic species from a different host [8,10,11,12,13,41,42,43]. *Diaporthe unshiuensis* was also reported as an endophyte of *Citrus* in China [13]. It was reported as a dieback disease associated with *Carya illinoinensisin* [71], peach [72], and grapevine [12] in China and as *Citrus* disease in Europe [7]. Therefore, further studies are necessary to understand the pathogenicity of these endophytic strains and the exact role of endophytic fungal taxa in the medicinal properties of these host plants.

In conclusion, in the present study, 12 endophytic species *Diaporthe* species were isolated from a traditional medicinal plant in China. To delineate these isolated taxa employing polyphasic approaches are necessary. Either morphology alone or phylogeny is difficult to delineate species in *Diaporthe*. Fungal growth media might have a significant effect on the number of different species isolated. Moreover, species occurrence is varied based on the plant tissue type as well. All previously known species obtained in this study have been reported as pathogens on various hosts in China and other countries. Therefore, future studies are necessary to understand the pathogenicity of these species on *M. officinalis*. The present study will be a baseline to understand the endophytic fungal diversity of Chinese Traditional Medicinal plants and thus to understand the effects of endophytes on medical properties.

## Figures and Tables

**Figure 1 jof-08-00806-f001:**
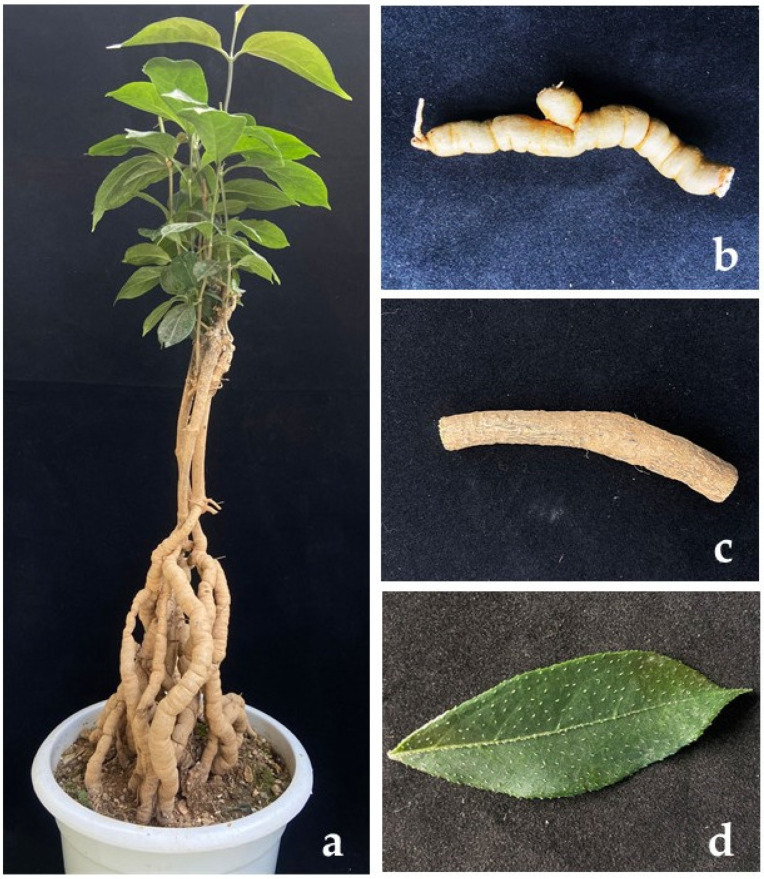
*Morinda officinalis* F.C. How. (*Rutaceae*) (**a**). The whole tree. (**b**). The root of the tree. (**c**). The stem of the tree. (**d**). The leaf of the tree.

**Figure 2 jof-08-00806-f002:**
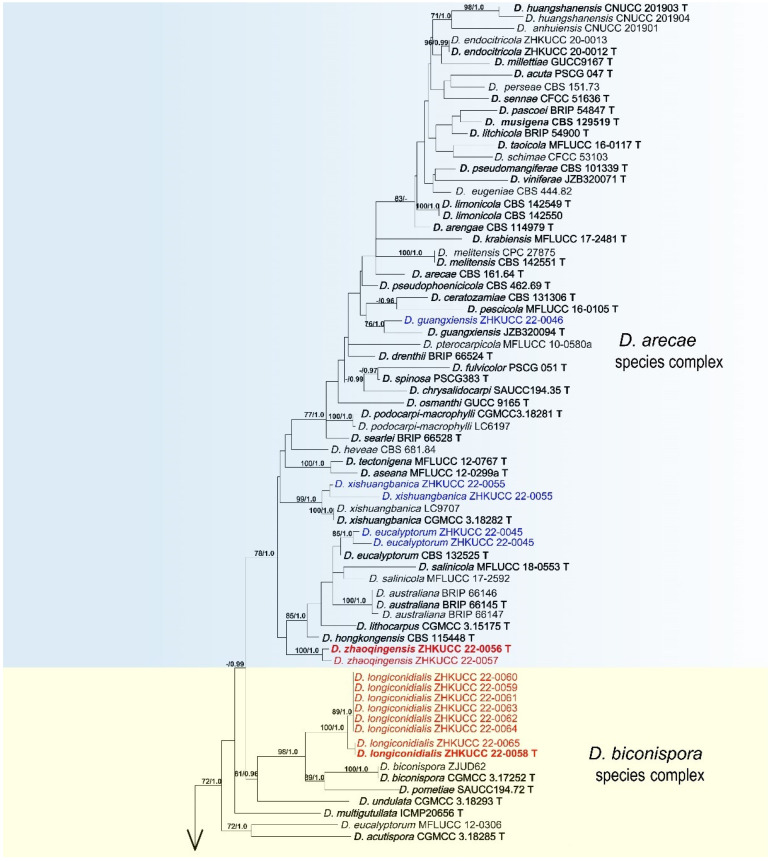
The best scoring RAxML tree obtained using the combined dataset of ITS, *tef1-α*, *cal*, *his*, and *β-tubulin* sequences. *Diaporthella corylina* (CBS 121124) was used to root the tree. Bootstrap support values equal to or greater than 60% in ML and BYPP equal to or greater than 0.95 are shown as ML/BYPP above the respective node. The isolates belonging to the current study are given in blue for known species, and novel taxa are shown in red. Ex-type strains are bold, with T at the end of the strains numbers. The expected number of nucleotide substitutions per site is represented by the scale bar.

**Figure 3 jof-08-00806-f003:**
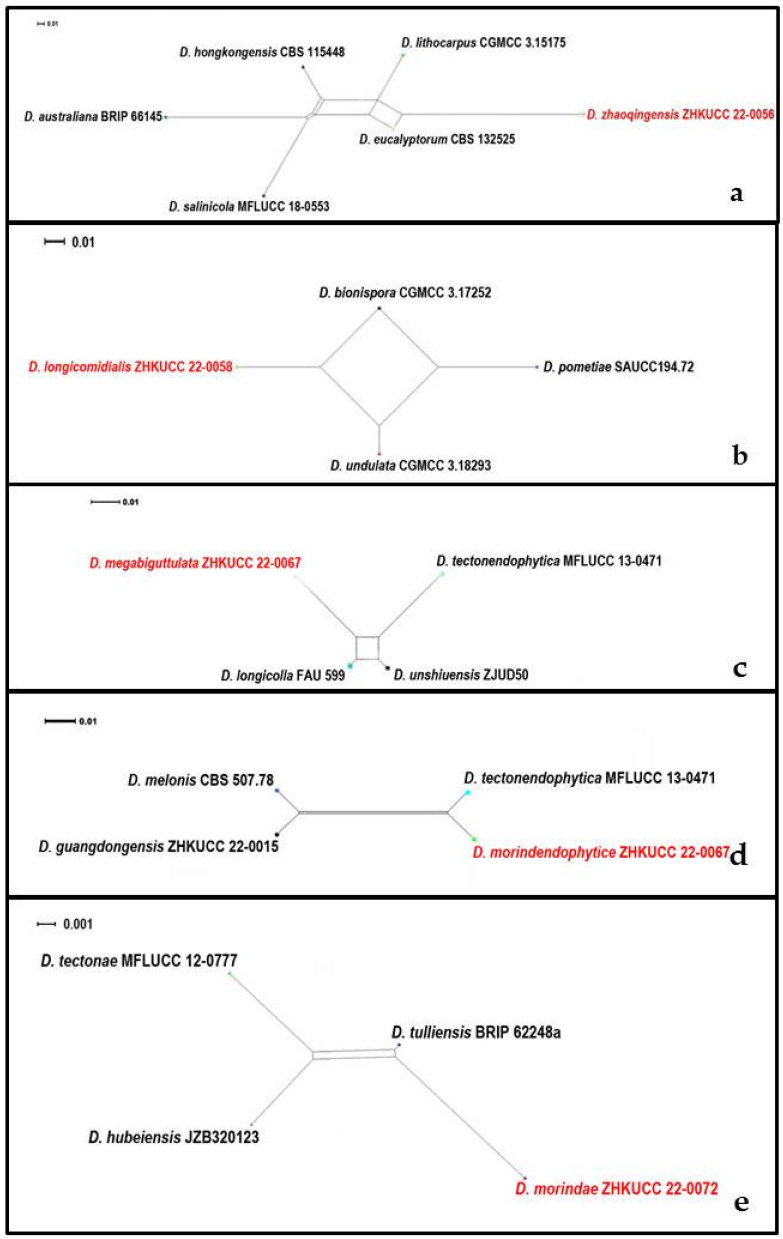
Split graphs showing the results of Pairwise homoplasy index test of (**a**) *Diaporthe zhaoqingensis* (Fw = 0.550), (**b**) *Diaporthe longiconidialis* (Fw = 0.300), (**c**) *Diaporthe megabiguttulata* (Fw = 1.000), (**d**) *Diaporthe morindendophytica* (Fw = 0.205), (**e**) *Diaporthe morindae* (Fw = 0.238) with their most closely related species using Log-Det transformation and splits decomposition options. The new taxon in each graph is shown in bold font.

**Figure 4 jof-08-00806-f004:**
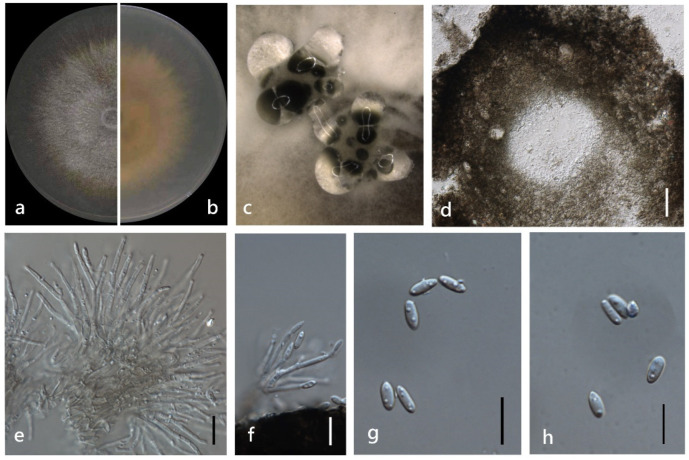
*Diaporthe chongqingensis* (ZHKUCC 22-0043) (**a**) upper view of colonies on PDA; (**b**) Reverse view of colonies on PDA; (**c**) Pycnidia with conidial droplets on PDA; (**d**) a vertical Section through a pycnidia; (**e**,**f**) Conidiogenous cells; (**g**,**h**) Alpha conidia. Scale bars: (**d**) =100 µm; (**e**–**h**) =10 µm.

**Figure 5 jof-08-00806-f005:**
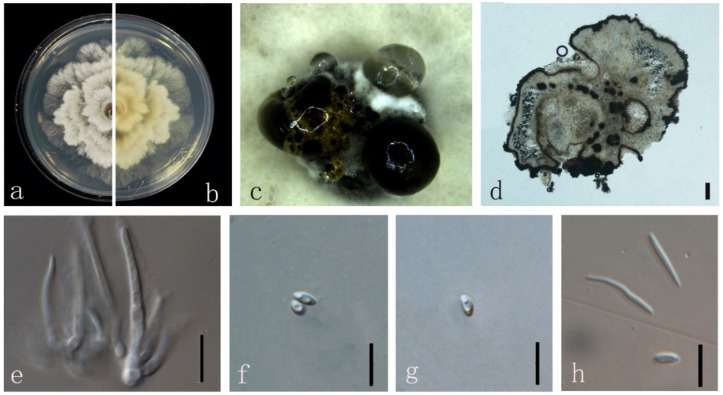
*Diaporthe eucalyptorum* (ZHKUCC 22-0044) (**a**) upper view of colonies on PDA; (**b**) Reverse view of colonies on PDA; (**c**) Pycnidia with conidial droplets on PDA; (**d**) Section of pycnidia; (**e**) Conidiogenous cells; (**f**–**h**) Alpha conidia; (**h**) Beta conidia. Scale bars: (**d**) =100 µm; (**e**–**h**) =10 µm.

**Figure 6 jof-08-00806-f006:**
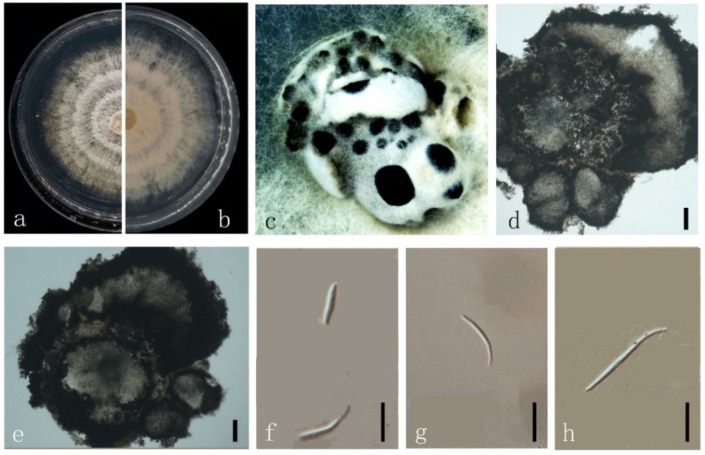
*Diaporthe guangxiensis* (ZHKUCC 22-0046) (**a**) upper view of colonies on PDA; (**b**) Reverse view of colonies on PDA; (**c**) Pycnidia with conidial droplets on PDA; (**d**,**e**) cross section of pycnidia; (**f**–**h**) Beta conidia; Scale bars: (**d**,**e**) =100 µm; (**f**–**h**) =10 µm.

**Figure 7 jof-08-00806-f007:**
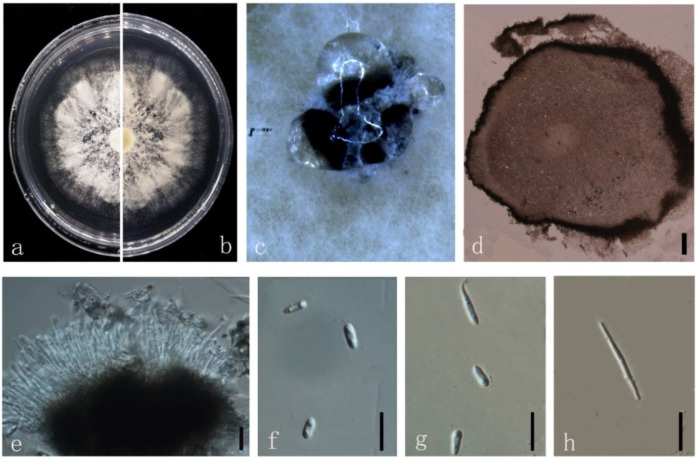
*Diaporthe heliconiae* (ZHKUCC 22-0047) (**a**) upper view of colonies on PDA; (**b**) Reverse view of colonies on PDA; (**c**) Pycnidia with conidial droplets on PDA; (**d**) vertical section of pycnidia; (**e**) Conidiogenous cells; (**f**,**g**) Alpha conidia; (**h**) Beta conidia. Scale bars: (**d**,**e**) =100 µm; (**f**–**h**) =10 µm.

**Figure 8 jof-08-00806-f008:**
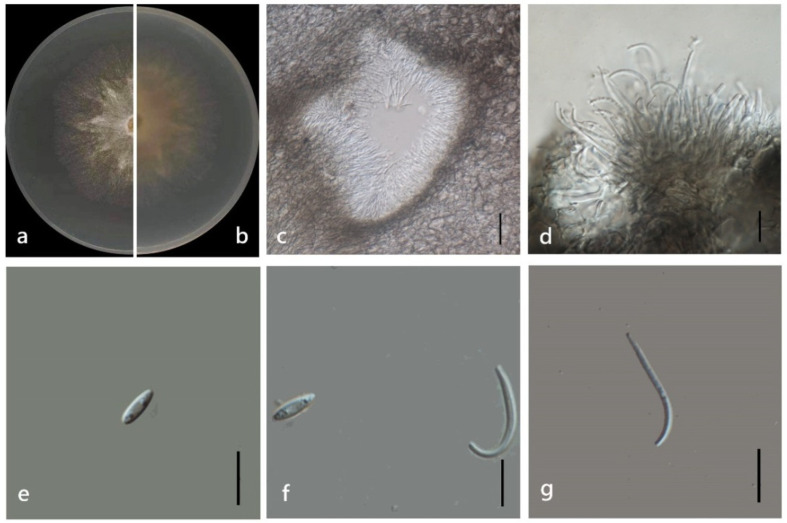
*Diaporthe longiconidialis* (ZHKUCC 22-0058) (**a**) upper view of colonies on PDA; (**b**) Reverse view of colonies on PDA; (**c**) Horizontal section through a pycnidia; (**d**) Conidiogenous cells; (**e**,**f**) Alpha conidia; (**f**,**g**) Beta conidia. Scale bars: (**c**) =100 µm; (**d**–**g**) =10 µm.

**Figure 9 jof-08-00806-f009:**
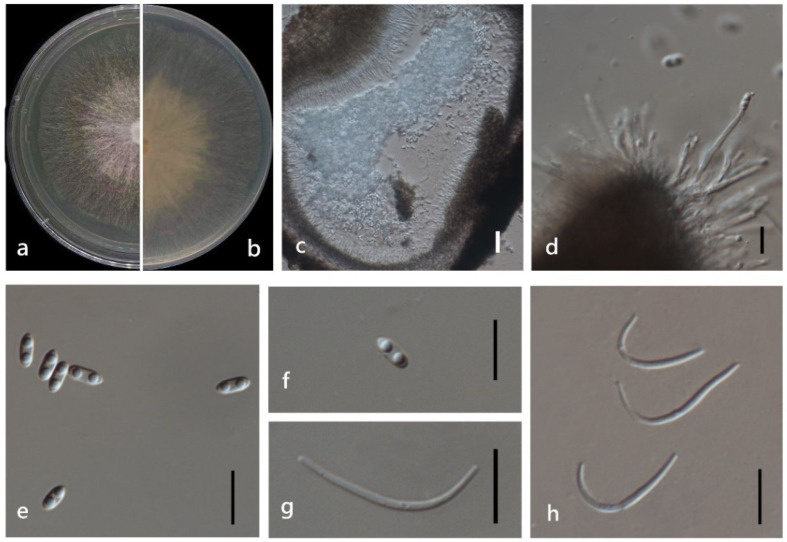
*Diaporthe megabiguttulata* (ZHKUCC 22-0067) (**a**) upper view of colonies on PDA; (**b**) Reverse view of colonies on PDA; (**c**) Horizontal section through a pycnidia; (**d**) Conidiogenous cells; (**e**,**f**) Alpha conidia; (**g**,**h**) Beta conidia. Scale bars: (**c**) =100 µm; (**d**–**h**) =10 µm.

**Figure 10 jof-08-00806-f010:**
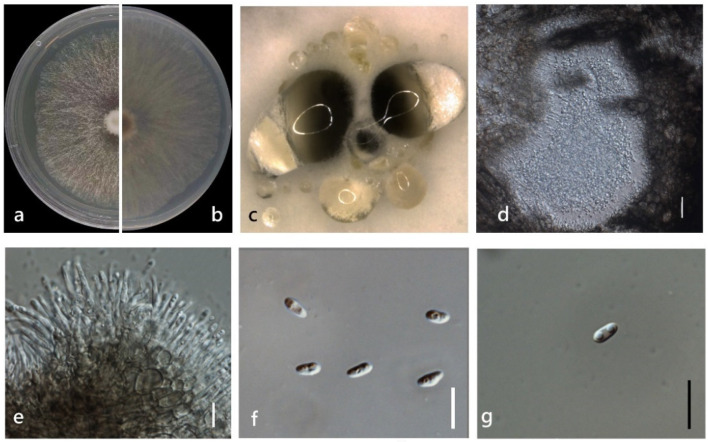
*Diaporthe morindae* (ZHKUCC 22-0072) (**a**) upper view of colonies on PDA; (**b**) Reverse view of colonies on PDA; (**c**) Conidiomata sporulating on PDA; (**d**) longitudinal section of Pycnidium; (**e**) Conidiogenous cells; (**f**,**g**) Alpha conidia. Scale bars: (**d**) =100 µm; (**e**–**g**) =10 µm.

**Figure 11 jof-08-00806-f011:**
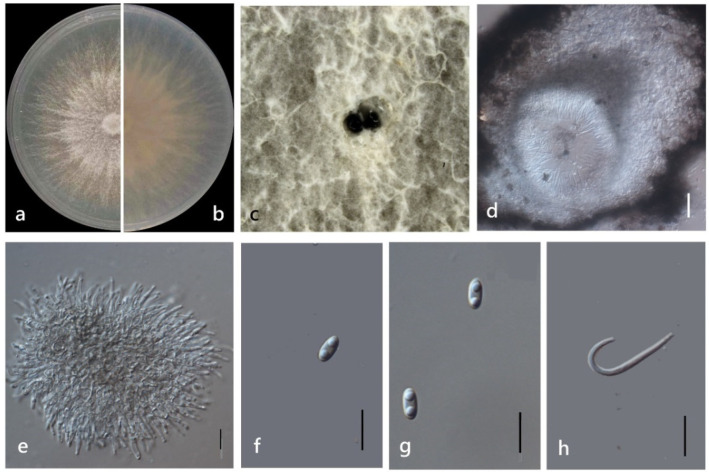
*Diaporthe morindendophytica* (ZHKUCC 22-0069) (**a**) Upper view of colonies on PDA; (**b**) Reverse view of colonies on PDA; (**c**) Pycnidia sporulating on PDA; (**d**) Section of pycnidia; (**e**) Conidiogenous cells; (**f**,**g**) Alpha conidia; (**h**) Beta conidia. Scale bars: (**d**) =100 µm; (**e**–**h**) =10 µm.

**Figure 12 jof-08-00806-f012:**
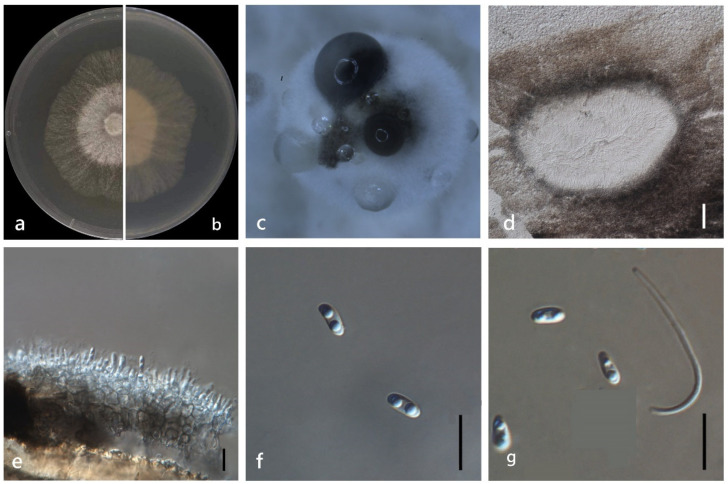
*Diaporthe siamensis* (ZHKUCC 22-0049) (**a**) upper view of colonies on PDA; (**b**) Reverse view of colonies on PDA; (**c**) Pycnidia with conidial droplets on PDA; (**d**) Section of pycnidia; (**e**) Conidiogenous cells; (**f**,**g**) Alpha and beta conidia. Scale bars: (**d**) =100 µm; (**e**–**g**) =10 µm.

**Figure 13 jof-08-00806-f013:**
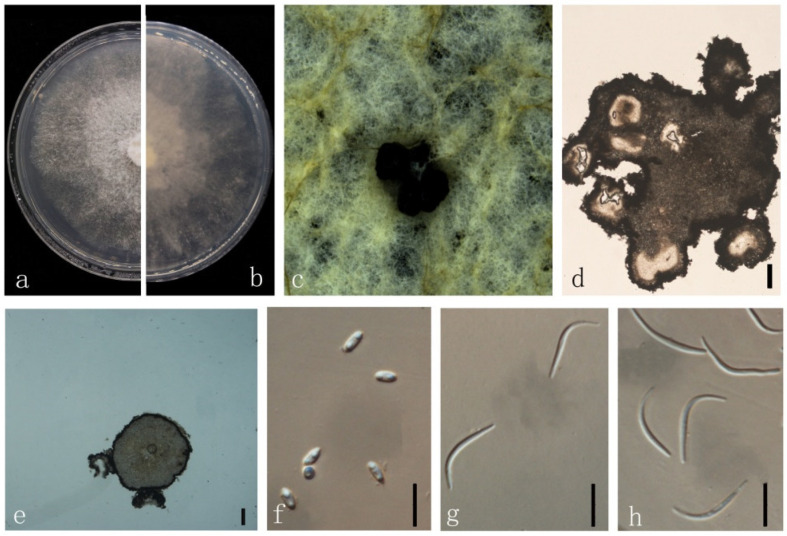
*Diaporthe unshiuensis* (ZHKUCC 22-0051) (**a**) upper view of colonies on PDA; (**b**) Reverse view of colonies on PDA; (**c**) Conidiomata sporulating on PDA; (**d**,**e**) Section of pycnidia; (**f**) Alpha conidia; (**g**,**h**) Beta conidia. Scale bars: (**d**,**e**) =100 µm; (**f**–**h**) =10 µm.

**Figure 14 jof-08-00806-f014:**
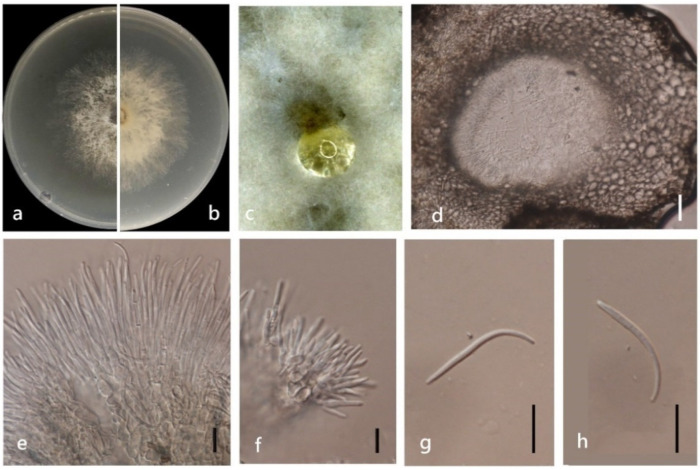
*Diaporthe xishuangbanica* (ZHKUCC 22-0054) (**a**) upper view of colonies on PDA; (**b**) Reverse view of colonies on PDA; (**c**) Pycnidia with conidial droplets on PDA; (**d**) Section of pycnidia; (**e**,**f**) Conidiogenous cells; (**g**,**h**) Beta conidia. Scale bars: (**d**) =100 µm; (**e**–**h**) =10 µm.

**Figure 15 jof-08-00806-f015:**
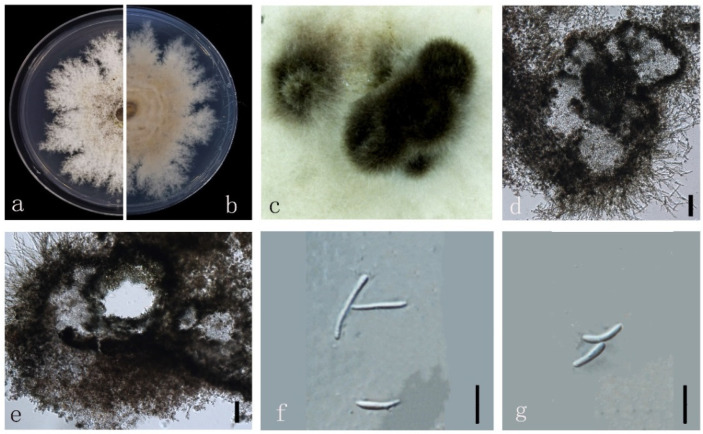
*Diaporthe zhaoqingensis* (ZHKUCC 22-0056) (**a**) upper view of colonies on PDA; (**b**) Reverse view of colonies on PDA; (**c**) Pycnidia with conidial droplets on PDA; (**d**,**e**) Pycnidium; (**f**,**g**) gamma conidia; Scale bars: (**d**,**e**) =100 µm; (**f**,**g**) =10 µm.

**Table 1 jof-08-00806-t001:** Gene regions, respective primer pairs, processes, and references used in the study.

Gene Region	Primers	Optimized PCR Protocols	References
ITS	ITS1	(94 °C: 30 s, 55 °C: 50 s, 72 °C: 1 min) × 35 cycles	[22]
	ITS4		
*tef1-α*	EF1-728F	(95 °C: 30 s, 58 °C: 30 s, 72 °C: 1 min) × 35 cycles	[23]
	EF1-986R		
*cal*	CAL-228F	(95 °C: 30 s, 55 °C: 50 s, 72 °C: 1 min) × 35 cycles	[23]
	CAL-737R		
*his*	CYLH4F	(95 °C: 30 s, 58 °C: 30 s, 72 °C: 1 min) × 35 cycles	[24]
	H3-1b		[25]
*β-tubulin*	BT2a	(94 °C: 30 s, 58 °C: 50 s, 72 °C: 1 min) × 35 cycles	[24]
	Bt2b		

## Data Availability

The sequence data generated in this study are deposited in NCBI GenBank (https://www.ncbi.nlm.nih.gov/genbank, accessed on 1 April 2022). All accession numbers are given in Appendix A.

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
