# Peer review of "Endophytic Diaporthe Associated with Morinda officinalis in China"

_jof, 2022, doi:10.3390/jof8080806_

Round 1

Reviewer 1 Report

This is an excellent fungal survey and follow-up systematics. However I identified some major revisions needed before publiscation. 

The Bayesian analysis was not run correctly, so I have to reject this manuscript and encourage resubmission. I also have serious doubts about the species determinations of some of the species you found. Your isolates of D. guangxiensis, D. xishuangbanica, D. heliconiae, D. chongqingensis and D. siamensis are all look quite genetically distinct from the other strains in their species - your phylogeny does not support your species determinations for these isolates, in my opinion. It is not clear from the manuscript what criteria was used to determine these isolates to be pre-existing species rather than describing them as novelties. It seems to me that these weren't described because they were not as commonly isolated, not because they were genetically closer to described species. For example, D. morindae is genetically closer to D. tectonae than your D. siamensis strains are to the conspecific MFLUCC strains you included in the phylogeny. This discrepancy needs to be addressed in a revision.   

39 italicize Phomopsis

40 this is the current recommended method for species determination, but there's nothing stopping researchers from doing morphological determinations

40-47 I agree with this paragraph but there may be a more direct way of stating it. "Polyphasic taxonomy has been recently employed for Diaporthe systematics, but, like many fungi there is no consistent criteria for delineating species" 

53-57 I think it is worth being explicit that the same species are found as pathogens, endophytes and saprotrophs on different hosts (ie, the "pogo stick hypothesis"). Currently it could be read that some species of Diaporthe are endophytes and others are pathogens. furthermore, I think you could use this paragraph to define endophyte as you use it in this particular manuscript - the term is widely used in different contexts, and because it is so central to this manuscript it should be defined within. 

79 collects -> collected

85-87 some of these should have recipes or citations

104 I think you should list the specific primer pairs, or cite Table 2 here

104-105 list PCR conditions

112 leave -> leaf

122 list whether phylogenetic were partitioned, and whether each subset was tested separately. If loci weren't partitioned during analysis, you must perform a partition-homogeity test to assess whether they can be concatenated. 

168-171 this is unnecessary data to include, especially if you're not going to discuss it

125 bootstrapping -> bootstrap 

127 You are listing unnecessary information about your Bayesian phylogeny, and omitting critical info: how did you determine that your two independent "runs" converged? 100,000 generations is not nearly enough for an analysis like this, you need at least two million, I would personally do at least 10 or 20 million. Critically, I would run them until I was sure that the runs had converged, and list the standard deviation of split frequencies. 

Author Response

Dear Reviewer

Thank you very much for your reviewer comments. We have corrected your manuscript following your comments and all changes are highlighted. We would like to answer your specific comments as below.

1: The Bayesian analysis was not run correctly, so I have to reject this manuscript and encourage resubmission. I also have serious doubts about the species determinations of some of the species you found. Your isolates of D. guangxiensis, D. xishuangbanica, D. heliconiae, D. chongqingensis and D. siamensis are all look quite genetically distinct from the other strains in their species - your phylogeny does not support your species determinations for these isolates, in my opinion. It is not clear from the manuscript what criteria was used to determine these isolates to be pre-existing species rather than describing them as novelties. It seems to me that these weren't described because they were not as commonly isolated, not because they were genetically closer to described species. For example, D. morindae is genetically closer to D. tectonae than your D. siamensis strains are to the conspecific MFLUCC strains you included in the phylogeny. This discrepancy needs to be addressed in a revision.   

Response: Thank you very much for your insightful comments on our manuscript. We are sorry for the mistake done in the Bayesian analysis. We have already run it correctly, however, it seems while we are proofreading the final version, we have messed up it with an old version. We are sorry for this. About the species delineation in Diapothe is always controversial. Diaporthe being and cryptic species results in many phylogenetic species over the last few years. Yet there is no study to either define these genotypes or to synonyms for many taxa as you mentioned because they are genetically closer. Because of this reason, we define several species thinking of them as already known species, we might need to apply the same rule to the whole genus and address this issue. Following these instincts yes we did focus to introduce them as new species based on recent studies. However, since you have pointed out this matter, we added a few sentences in the discussion mentioning, that we need quick solutions to the enigma in this genus.

3: 40 this is the current recommended method for species determination, but there's nothing stopping researchers from doing morphological determinations

Response: Thank you very much for your comments. Yes, we shouldn’t ignore morphology at all. When it comes to cryptic genera like Diaporthe, morphology alone can not help.

4: 40-47 I agree with this paragraph but there may be a more direct way of stating it. "Polyphasic taxonomy has been recently employed for Diaporthe systematics, but, like many fungi there is no consistent criteria for delineating species" 

Response: Thank you very much for this suggestion, therefore we conclude our sentence following your suggestion.

5: 53-57 I think it is worth being explicit that the same species are found as pathogens, endophytes and saprotrophs on different hosts (ie, the "pogo stick hypothesis"). Currently it could be read that some species of Diaporthe are endophytes and others are pathogens. furthermore, I think you could use this paragraph to define endophyte as you use it in this particular manuscript - the term is widely used in different contexts, and because it is so central to this manuscript it should be defined within. 

Response: Thank you very much for this. We rearranged the section following your comments hope you agree with the changes.

11: 122 list whether phylogenetic were partitioned, and whether each subset was tested separately. If loci weren't partitioned during analysis, you must perform a partition-homogeity test to assess whether they can be concatenated. 

Response: Thank you very much for this. For each gene region, we did the Mr model test separately. The model chosen for each gene region by the Mr model analysis was GTR + I +G. Then we partitioned the tree and finalized the analysis. However, since you have mentioned we did the homogeneity test and the results suggested that all gene regions can be concatenated.

12: 168-171 this is unnecessary data to include, especially if you're not going to discuss it

Response: Thank you very much for this. We keep it as it is since this is the result and we discuss those we concerned.

14: 127 You are listing unnecessary information about your Bayesian phylogeny, and omitting critical info: how did you determine that your two independent "runs" converged? 100,000 generations is not nearly enough for an analysis like this, you need at least two million, I would personally do at least 10 or 20 million. Critically, I would run them until I was sure that the runs had converged, and list the standard deviation of split frequencies. 

Response: Thank you very much for this. We are sorry for the mistake here. We apologise for the mistake as I explained in the first comment.

Reviewer 2 Report

Dear Authors

The manuscript entitled “Endophytic Diaporthe Associated with Morinda officinalis in China” is interesting. The manuscript overall is good and it is well-worth publishing. The content of the paper could be published in JoF. I think the study is good and the data presented in the manuscript is sufficient but I have noticed that there is need some minor improvement in the manuscript. In the manuscript, I pointed out a few minor corrections.

Best regards

Author Response

Dear Reviewer

Thank you very much for your comments. We have corrected the comments from you and all changes are highlighted. You can find from the paper.

Reviewer 3 Report

Dear Authors, 

please find below some minor comments.

Line 79-> ..”were collected”

Line 80-> ..”the collected samples were transported to the laboratory where were subjected to isolation of endophytic strains procedure” (or similarly..)

Line 377: revise figure 8 legend

Line 383: “spores”

Line 455: revise figure 10 legend

Line 526: revise figure 12 legend

Line 535: ..”consisting of several layers”… (please check “sevea” throughout the text)

Line 628: revise figure 15 legend

Line 700: this is not a conclusion, but a result…

Line 704: “leaves”

 Congratulations!

Author Response

Dear Reviewer

Thank you very much for your reviewer comments. We have corrected the minor comments from you and all changes are highlighted. However, we would like to have one more clarification.

Line 79-> ..”were collected”

Response: Thank you very much for this. We have corrected it.

Line 80-> ..”the collected samples were transported to the laboratory where were subjected to isolation of endophytic strains procedure” (or similarly..)

Response: Thank you very much for this. We have revised this in the paper.

Line 377: revise figure 8 legend

Response: Thank you very much for this. We have corrected it.

Line 383: “spores”

Response: I am so sorry about this and thank you very much for this. We have corrected it.

Line 455: revise figure 10 legend

Response: I am so sorry we made a mistake. We have corrected it.

Line 526: revise figure 12 legend

Response: Thank you very much for this. We have corrected it.

Line 535: ..”consisting of several layers”… (please check “sevea” throughout the text)

Response: I am so sorry we made a mistake. We have corrected it.

Line 628: revise figure 15 legend

Response: Thank you very much for this. We have corrected it.

Line 700: this is not a conclusion, but a result…

Response: Thank you for your comments. We have revised and highlighted in the article.

Line 704: “leaves”

Response: Thank you very much for this. We have corrected it.

Round 2

Reviewer 1 Report

I did not find many of my previous issues were adequately addressed in the manuscript despite the statements made in the response. Unfortunately most of the new edits introduced grammatical errors. 

None of the very important issues regarding the inconsistent species determinations were considered.

You must state that your Bayesian runs converged and the maximum standard deviation of split frequencies, as was previously requested.

Because of the inconsistency with which their methodology was described without changing the resulting figures, I do not trust the accuracy phylogenetic results that are presented. Because no efforts were made to improve the analysis, I will now recommend rejection without encouragement to resubmit. 

41-43 in most cases it is not the type specimens that help establish the species boundaries, but non-type, voucher specimens, often identified by the same author who described the species or monographed the genus. I would restate it as "type and voucher specimens" 

43-44 this sentence is not complete

46-48 the paper you cited did not use GCPSR to better delineate species within a complex - their results found that the complex should be reduced to a single species (I also reviewed that manuscript). 

59-61 this sentence needs grammar improvement. Additionally, the world of endophytes is far, far more diverse than the genera you mention. Highly recommend citing doi:  10.1071/AP05082

Author Response

Thank you very much for your comments. We are sorry that we did not re-run the tree. One reason was out time was not enough to re-run the tree with big data set. However, following the reviewer’s comment, we rearrange the phylogenetic analysis. In which we run the tree with partitions and apply different evolutionary models for each gene region following J-Model analysis. Here we first obtained the ML tree, and they find the species complexes for our isolates. Then adding only taxa belonging to five species complexes we developed the tree. We have corrected your manuscript following the comments. We would like to answer your specific comments as below.

Point 1: 41-43 in most cases it is not the type specimens that help establish the species boundaries, but non-type, voucher specimens, often identified by the same author who described the species or monographed the genus. I would restate it as "type and voucher specimens" 

Response 1: Thank you very much for your comments. We do agree with you and have revised the article.

Point 2: 43-44 this sentence is not complete

Response 2: Thank you very much for your comments. I am so sorry about this. And we revised it.

Point 3: 46-48 the paper you cited did not use GCPSR to better delineate species within a complex - their results found that the complex should be reduced to a single species (I also reviewed that manuscript). 

Response 3: Thank you very much for your comments. I am so sorry about this. We have updated the reference.

Point 4: 59-61 this sentence needs grammar improvement. Additionally, the world of endophytes is far, far more diverse than the genera you mention. Highly recommend citing doi:  10.1071/AP05082

Response 3: Thank you for your comment. We have added the reference and revised the expression about it.
